# Combined Pharmacological Conditioning of Endothelial Cells for Improved Vascular Graft Endothelialization

**DOI:** 10.3390/ijms26157183

**Published:** 2025-07-25

**Authors:** Zhiyao Lu, Xuqian Zhou, Xiaowen Liu, Chunyan Liu, Junfeng Zhang, Lei Dong

**Affiliations:** 1State Key Laboratory of Pharmaceutical Biotechnology, School of Life Sciences, Nanjing University, 163 Xianlin Avenue, Nanjing 210023, China; dz1830003@smail.nju.edu.cn (Z.L.); azhouxuqian@163.com (X.Z.);; 2Wuxi Xishan NJU Institute of Applied Biotechnology, Wuxi 214000, China; 3Chemistry and Biomedicine Innovation Center, Nanjing University, Nanjing 210093, China

**Keywords:** endothelialization, human umbilical vein endothelial cells (HUVEC), endothelial cell heterogeneity, tight junction, pharmacological regulation, anticoagulation

## Abstract

The development of functional endothelial monolayers on synthetic vascular grafts remains challenging, particularly for small-diameter vessels (<6 mm) prone to thrombosis. Here, we present a pharmacological strategy combining 8-(4-chlorophenylthio) adenosine 3′,5′-cyclic monophosphate sodium salt (pCPT-cAMP, a tight junction promoter) with nitric oxide/cGMP pathway agonists 3-morpholinosydnonimine (SIN-1), captopril, and sildenafil) to enhance endothelialization. In human umbilical vein endothelial cells (HUVECs), this four-agent cocktail induced a flat, extended phenotype with a 3-fold increased cell area and 57.5% fewer cells required for surface coverage compared to controls. Immunofluorescence analysis revealed enhanced ZO-1 expression and continuous tight junction formation, while sustained nitric oxide (NO) production (3.9-fold increase) and restored prostacyclin (PGI_2_) secretion demonstrated preserved endothelial functionality. Anticoagulation assays confirmed a significant reduction in thrombus formation (*p* < 0.01) via dual inhibition of platelet activation and thrombin binding. These findings establish a synergistic drug combination that promotes rapid endothelialization while maintaining antithrombogenic activity, offering a promising solution for small-diameter vascular grafts. Further studies should validate long-term stability and translational potential in preclinical models.

## 1. Introduction

Vascular replacement surgeries are frequently performed in clinical practice. While autologous grafts harvested from donor sites, such as the saphenous vein, remain the gold standard, they can only fulfill < 50% of clinical demand [1]. Synthetic vascular prosthetics have been widely adopted as alternatives. Although large-diameter grafts (>6 mm) demonstrate satisfactory performance, small-diameter grafts (<6 mm) are prone to thrombosis and intimal hyperplasia, leading to severe cardiovascular complications [2].

While synthetic vascular prosthetics provide structural supplementation, their lack of functional endothelial monolayers remains a critical limitation. The intact endothelium plays an indispensable role in maintaining vascular homeostasis through its non-thrombogenic properties [3,4]. Although material surface modifications can enhance endothelial cell (EC) adhesion, most approaches fail to achieve complete surface coverage and preserve the anticoagulant bioactivity of seeded ECs, primarily due to structural and functional deficiencies of the cultured cells [5].

Vascular ECs maintain anticoagulation through the following two primary mechanisms: (1) expression of anticoagulant molecules, particularly prostacyclin (PGI_2_) and nitric oxide (NO), which synergistically inhibit platelet activation and aggregation through cAMP/PKA and cGMP/PKG signaling, respectively; and (2) maintenance of endothelial integrity to prevent tissue factor exposure [6,7]. A continuous, non-leaky endothelial layer is essential to prevent exposure of tissue factors to circulating coagulation factors [8,9]. It is the tight junction between adjacent cells that serves the major functional purpose of providing a barrier [10,11]. However, cultured ECs often exhibit disrupted junctional complexes. Previous studies have demonstrated that hydrocortisone, cAMP, and its derivatives can strengthen these intercellular connections [12,13,14,15,16].

Lineage-tracing studies have revealed that during vasculature development, endothelial cells migrate in a vein-to-capillary-to artery direction as the vascular network expands and matures. This indicates vein endothelial cells as a possible source to proliferate and differentiate into a vascular endothelial cell pool, which facilitates in vivo engineering [17,18,19,20]. Meanwhile, the phenotypic plasticity during culture challenges the potency of pro-endothelialization treatment, making vein endothelial cells an ideal cell model, among which the intensively studied umbilical vein endothelial cell (HUVEC) provides the most abundant literature for reference. Morphological changes can lead to the dominance of specific phenotypes, potentially disrupting cell–cell connections through mechanical forces [21]. A flat and extended phenotype facilitates proliferation and migration, while the rounded phenotype disrupts monolayer integrity and tight junction formation [22,23,24,25].

Given the limitations of gene editing in normal cells, we aimed to develop a pharmacological approach to establish an anticoagulant, tightly connected endothelial monolayer by enhancing cAMP/PKA and cGMP/PKG signaling pathways. This study presents a novel combination of agents that promotes the induction and maintenance of an extended endothelial phenotype in HUVEC cultures. Our treatment regimen demonstrated significant improvements in EC spreading and tight junction formation. Moreover, the resulting monolayer exhibited robust NO production, which is crucial for vascular endothelial stability and anticoagulant function in vivo, suggesting promising potential for effective endothelialization.

## 2. Results

### 2.1. 3-Morpholinosydnonimine (SIN-1) Promotes Endothelial Cell Growth Through Nitric Oxide (NO) Delivery

NO plays a pivotal role in regulating vascular endothelial bioactivity and function. As the primary mediator of vasodilation, NO activates sGC, stimulating cGMP synthesis, which induces smooth muscle cell relaxation and inhibits platelet aggregation. During angiogenesis, NO promotes EC growth and migration while protecting against apoptosis, thereby maintaining vascular endothelial integrity [26]. Given that optimal endothelialization requires EC proliferation and surface coverage, NO supplementation may enhance these processes. Among the commonly used exogenous NO donors in clinical and experimental settings, we evaluated sodium nitroprusside (SNP), 3-morpholinosydnonimine (SIN-1), and S-nitroso-N-acetylpenicillamine (SNAP) to identify the most suitable additive for endothelial cell culture [27].

HUVECs were cultured in complete medium supplemented with SNP (100 μM), SNAP (1 μM), SIN-1 (100 μM), or PBS as a control [7,28,29,30,31,32]. Endothelial cell heterogeneity was most evident at the morphological level. Control cells cultured in standard medium initially exhibited a characteristic cobblestone morphology, which became progressively disorganized over time. After 24 h of static culture, cellular morphology was analyzed using a segmentation algorithm (Cellpose v2.2.2) to quantify cell boundaries (Figure 1A). We performed concentration gradients, and SNAP treatment at low concentrations induced widespread cell death, demonstrating significant cytotoxicity. SNP treatment showed minimal effects, with only a slight increase in the proportion of small-sized cells (Figure 1B). In contrast, SIN-1 treatment enhanced HUVEC viability, supporting larger cell sizes and improved morphological uniformity compared to other NO donors (Figure 1B,C).

### 2.2. Pharmacological Modulation of Endothelial Cell Activity Promotes Favorable Morphological Changes

Beyond exogenous NO supplementation, endogenous NO production through endothelial nitric oxide synthase (eNOS) activation represents another strategy to enhance the NO/cGMP pathway. Angiotensin II (Ang II) suppresses both eNOS expression and NO generation [33]. Captopril is an angiotensin-converting enzyme (ACE) inhibitor that reduces Ang II levels while potentiating bradykinin, thereby enhancing NO and PGI_2_ production [34]. Sildenafil, a phosphodiesterase-5 (PDE5) inhibitor, prevents cGMP degradation, prolonging NO-mediated effects. Both captopril and sildenafil provide additional antioxidant protection for ECs, making them promising candidates for endothelialization enhancement [35,36].

While EC proliferation and spreading are essential for endothelialization, the formation of intercellular tight junctions is equally critical for functional monolayer establishment. Glucocorticoids and cAMP analogs have been shown to strengthen tight junctions [37,38]. We selected representative agents from each class. Captopril (0.3 mM) was also included due to its ability to stabilize endothelial barriers by counteracting Ang II-mediated tight junction disruption [39]. The concentrations were chosen by concentration gradient experiments, referring to previous reports [40,41]. The reported concentrations were generally proved valid and adopted, except for 8-(4-chlorophenylthio) adenosine 3′,5′-cyclic monophosphate sodium salt (pCPT-cAMP), which better enhanced the expression of ZO-1 at a higher concentration than reported (0.25 mg/mL) [14] (Appendix A).

Morphological analysis revealed distinct responses to different treatments (Figure 2A–C). Hydrocortisone did not significantly alter cell size or shape compared to controls. Interestingly, cells treated with either captopril or sildenafil (10 μM) exhibited similar morphological characteristics to those treated with pCPT-cAMP, showing improved cell–cell connections with less defined boundaries and reduced elongation, suggesting better phenotypic maintenance. Quantitative analysis showed no significant differences in cell area or perimeter among these treatment groups (Figure 2B,C), but qualitative observations indicated enhanced intercellular connectivity.

### 2.3. Synergistic Effects of Agent Combination Enhance Endothelial Cell Spreading

Based on the individual effects observed, we investigated whether combining SIN-1, captopril, sildenafil, and pCPT-cAMP could produce synergistic benefits for endothelialization. At low magnification, cells treated with SIN-1 + sildenafil + captopril or the complete four-agent combination exhibited directional alignment, whereas other treatment groups showed random clustering (Figure 3A). Higher magnification revealed the following distinct morphological differences: cells treated with sildenafil + captopril or sildenafil + captopril + pCPT-cAMP maintained polygonal shapes similar to controls, with visible cell boundaries indicating surface repulsion (Figure 3A).

The complete four-agent combination (SIN-1 + sildenafil + captopril + pCPT-cAMP) induced a unique morphological pattern characterized by large, flat cells with extensive spreading and intercellular connections (Figure 3A). Quantitative analysis demonstrated significant increases in both cell area (almost 3-fold) and perimeter (over 1.2-fold) compared to controls (Figure 3B,C). Remarkably, this combination achieved complete surface coverage with 57.5% fewer cells than standard culture conditions (Figure 3A,D), suggesting enhanced spreading efficiency and potential for rapid endothelialization. Cytotoxicity assessments were performed on the four-component combination and individual compounds to determine optimal concentrations for each agent (Appendix A). The agents, particularly the full combination, exhibited a lower cell proliferation rate than expected, as microscope observations exhibited robust cell viability. Given the contact inhibition effect, this is considered to be associated with the increased area of individual cells.

### 2.4. Agent Combination Enhances Tight Junction Formation Beyond pCPT-cAMP Alone

Endothelial cell phenotype is mainly regulated by cytoskeletal organization and cell–cell adhesion [42,43]. Previous studies have demonstrated that cAMP and its analogs improve endothelial junction assembly and maintenance [14]. Our observations of the distinctive cell patterns induced by NO/cGMP pathway agonists suggested potential benefits for junction formation. Therefore, we examined the effects of our agent combinations on tight junction assembly using ZO-1 immunofluorescence staining (Figure 4A) [44].

Control cells showed discontinuous ZO-1 expression, while pCPT-cAMP treatment alone increased ZO-1 levels but failed to establish continuous junctions (Figure 4A,B). NO/cGMP pathway agonists promoted linear tight junction formation but induced cellular transformation, consistent with bright-field observations (Figure 4A). The complete four-agent combination not only preserved continuous junctions but also significantly expanded cell boundaries, with average cell area tripling and corresponding perimeter increasing (Figure 4A,D,E).

ZO-1 is believed to link the transmembrane protein occludin with the actin cytoskeleton [45]. A fluorescent phalloidin probe for actin revealed distinct organized patterns (Figure 4C,F) [42]. While pCPT-cAMP alone did not alter the contractile morphology, cGMP stimulation induced bipolar extension. The complete combination resulted in fully expanded cells with outward actin pulling, achieving sufficient surface coverage. Low-magnification observations suggested that elongated cytoskeletal proteins might contribute to more organized cellular arrangements on a larger scale (Figure 3A). In addition to ZO-1, tight junction-associated molecules include Claudins and JAMs. CLDN1 and CLDN5 are representative members of the Claudin family that function as sealing proteins. JAMs consist five members, among which JAM-A, JAM-B, and JAM-C are expressed by ECs. JAMs participate in the assembly of tight junctions and adhesions junctions. We performed qPCR analyses on these molecules and found that all of them were upregulated at the transcriptional level (Figure 4G). Surprisingly, ZO-1 showed the lowest increase. This indicates that rather than enhance expression, the agents may help stabilize ZO-1 protein. Although mRNA was upregulated, We did not see increased CLDN5 protein in the immunofluorescence assay. The phenomenon may involve downstream translation and protein–protein interactions. The combination of SIN-1+Cap+Sil+pCPT-cAMP greatly promoted the transcription of all six molecules.

To functionally evaluate the barrier integrity of the formed endothelial layer, we conducted a canonical permeability assay using Evans Blue and Transwell inserts. HUVECs were seeded on the inserts and cultured until full coverage. Evans Blue (0.67 mg/mL) was added into the inserts carpeted with HUVECs. The lower chambers were filled with 0.6 mL of 4% BSA to avoid leakage caused by the liquid level difference. After 1 h of balance in the cell incubator, the concentration of Evans Blue in the lower chamber was tested. pCPT-cAMP, SIN-1 + Cap + Sil, and the full combination all significantly inhibited the leakage of Evans Blue. Either SIN-1 + Cap + Sil or the four-component cocktail effectively reduced effusion by half (Figure 4H), demonstrating the functional barrier’s property.

The extensive upregulation of tight junction-forming molecules suggests activation of associated signaling pathways, such as the ANGPT1/TEK pathway, PI3K/AKT pathway, RHOA/ROCK pathway, and NF-κB and NOTCH1 pathway [46,47,48,49]. We gained a general view of the activity of these pathways by evaluating the transcriptional level of key regulators (Figure 4I). ANGPT1 exhibited the most significant increase in transcriptive activity, which was multiplied by over 200-fold with the four-component cocktail. The expression level of NOTCH1 increased by several times. The activation of these two signaling pathways promotes the expression and stability of ZO-1. AKT is another agonist facilitating the assembly of tight junctions in physiological condition, but it did not appear relevant here. The inflammatory pathway guided by NF-κB destroys tight junctions, the inhibition of which by the agents helps to maintain tight junction integrity. The active ANGPT1/TEK pathway also could play a role in suppressing NF-κB. Last but not least, the RHOA/ROCK pathway regulates the rearrangement of F-actin. If activated, the pathway will lead to cytoskeletal contraction and the collapse of tight junctions. The downregulation of RHOA contributes to an expanded cellular morphology and prevents the disruption of tight junctions. Together, the compounds enhanced the presence of tight junctions, promising ideal endothelialization.

### 2.5. Endothelial Monolayer Induced by Agent Combination Demonstrates Antithrombogenic Properties

Beyond structural integrity, maintenance of endothelial homeostasis through sustained NO and PGI_2_ production is essential for anticoagulation and smooth muscle cell proliferation regulation [4,50,51]. The synergistic action of NO and PGI_2_ enhances cGMP/cAMP-mediated platelet activation inhibition. We successively assessed NO and PGI_2_ production in culture supernatants over three days under different treatment conditions, until the cells covered the dish (Figure 5A,B).

The complete four-agent combination increased NO content by 3.9-fold, maintaining stable levels throughout the observation period (Figure 5A). Although SIN-1 hydrolysis might influence initial NO elevation, sustained NO levels indicated endogenous synthesis by ECs, potentially facilitated by pCPT-cAMP-mediated homeostasis. PGI_2_ secretion showed an initial decline followed by gradual recovery by 72 h (Figure 5B), possibly reflecting adaptive responses to the sudden NO increase.

Anticoagulation assays using human blood treated with platelet-rich plasma (PRP) demonstrated significant differences among the groups (Figure 5C,D). Interestingly, pCPT-cAMP alone exacerbated coagulation compared to controls. The complete combination created a NO-rich environment that effectively reduced thrombus formation. The lower EC density may correlate with reduced thrombin binding sites, partially explaining the observed antithrombogenic effects [52]. The evident effect might result from core events in the coagulation process. Tissue factor (F3) plays a key role in launching the downstream coagulation cascade reaction, and Von Willebrand factor (VWF), the key regulator in flow, promotes platelet aggregation. We evaluated the mRNA level of the two primary factors (Figure 5E). It is exciting to find that pCPT-cAMP alone boosted the transcription of F3, which might promote blood coagulation. This agrees with an increased thrombus weight (Figure 5D). The addition of 3 NO/cGMP agonists alleviated the effect. On the other hand, pCPT-cAMP, NO/cGMP agonists and the combination of them all inhibited the expression of VWF, which would prevent platelet aggregation and thrombotic events. Furthermore, the combination performed better than either of the components. Overall, the four-agent treatment guided the ECs in a non-thrombogenic fashion via platelet inhibition.

## 3. Discussion

The inherent heterogeneity of endothelial cells presents significant challenges for achieving effective endothelialization of synthetic vascular grafts. Inadequate endothelialization remains a major contributor to severe cardiovascular complications following graft implantation. In this study, we developed a novel pharmacological cocktail combining pCPT-cAMP, a tight junction promoter, with three NO/cGMP pathway agonists (SIN-1, captopril, and sildenafil) to address these limitations. Our approach induced distinct morphological and functional changes in HUVECs, including the establishment of a flat, extended phenotype and enhanced tight junction formation, facilitating rapid endothelial coverage.

The observed synergistic effects of our combination treatment represent a significant advancement over previous endothelialization strategies. While cAMP analogs have been shown to improve tight junction assembly, our findings demonstrate for the first time that NO and its pathway agonists can further enhance this process. The underlying mechanisms warrant further investigation but may involve cGMP-mediated cytoskeletal reorganization and junctional protein stabilization.

Notably, our treatment regimen achieved effective endothelialization at lower cell densities compared to conventional methods. This density reduction, combined with the observed morphological changes, may contribute to the enhanced antithrombogenic properties of the resulting endothelial monolayer. The maintenance of NO and PGI_2_ production, particularly the stable NO levels observed over three days, suggests preservation of critical endothelial functions. Nonetheless, the present results have only been achieved in HUVECs. The agents are expected to be tested in more EC types, particularly artery ECs, because the artery endothelium demonstrates stronger impermeability properties. Its natural response to high shear stress activates the eNOS/NO signaling pathway, benefiting long-term maintenance of the established endothelium. Also, artery ECs may better adapt to the implantation site, since it is usually a section of diseased artery that is replaced.

The potential clinical implications of our findings are substantial. The ability to rapidly establish a functional endothelial layer on synthetic grafts could significantly improve outcomes for patients requiring small-diameter vascular replacements. However, several considerations must be addressed in future studies. First, while our results demonstrate the promising short-term anticoagulation effect of a drug combination in static culture, the durability of these improvements requires evaluation in flow and in animal models. Sheer stress produced by laminar flow is believed to inhibit coagulation by preventing platelet aggregation. The mechanical force factor may cooperate with the pharmacological treatments, serving as the activator of eNOS in the long run. When applied in vivo, one problem is the compatibility between the in vitro endothelialized material and autologous vessels at the anastomotic site, where they can induce coagulation due to vascular injury. In addition, whether the artificial endothelium layer survives the clearance from the body’s immune system is another aspect to monitor, as it may impair the antithrombotic effect. Second, further investigation is needed to elucidate the molecular pathways underlying the observed synergistic effects, for instance, the interplay between cAMP and cGMP signaling in junctional complex formation [53]. Furthermore, more pathways are involved here, as shown by qPCR analyses. Their interplay with one another may produce synergistic or antagonistic effects, like the cAMP/PKA and RHOA/ROCK signaling pathways, which demands further exploring. Third, the safety and efficacy of this pharmacological approach must be validated in human-derived endothelial cells and preclinical models before clinical application. Future studies should assess the performance of this endothelialization strategy under disease-relevant conditions, such as hyperglycemia or hyperlipidemia, to better predict clinical utility.

Our findings contribute to the growing body of evidence supporting pharmacological approaches to endothelialization. The combination of established cardiovascular drugs with well-characterized safety profiles may facilitate faster clinical translation compared to gene-based or material modification strategies. Future research directions should focus on optimizing the treatment protocol, exploring alternative agent combinations, and developing delivery systems for localized application in vascular grafts.

## 4. Materials and Methods

### 4.1. Reagents and Antibodies

The key reagents and antibodies used in the study are listed below (Table 1).

### 4.2. Cell Culture

HUVECs (Cat#PCS-100-013) were obtained from ATCC (ATCC, Manassas, VA, USA) and tested to determine that they were pathogen free. Cells were maintained in vascular cell basal medium (ATCC, Cat#PCS-100-030) (ATCC, Manassas, VA, USA) with 1% penicillin–streptomycin solution (Bio-Channel, Cat#BC-CE-007) (Bio-Channel, Nanjing, China) in the incubator (37 °C, 5% CO_2_).

### 4.3. Cell Toxicity Assay

Cell toxicity was evaluated using a Cell Counting Kit-8 (CCK-8) (APExBIO, Houston, TX, USA). Cells were seeded into a 96-well plate at a density of 10^4^ cells/100 μL/well. For each group, we set up three replicates. After 24 h of standard culture, drugs were added. Cells were incubated for 12 h before tests. We added 10 μL of CCK-8 solution into each well, including the blank and control group. After another 4 h of incubation, absorbance at λ = 450 nm was measured by a multimode microplate reader (Thermo Scientific, Waltham, MA, USA).

### 4.4. Immunofluorescent Staining

Cells were fixed in 2% paraformaldehyde solution for 20 min and then washed twice with PBS. We blocked cells with 5% BSA (0.5% Triton-X plus) for 1 h at room temperature, and then incubated them with primary antibodies diluted in 1% BSA overnight at 4 °C. The next day, the slides were washed with PBS and incubated with secondary antibodies for 45 min at room temperature. Cells were washed again with PBS and dyed with DAPI. Images were acquired using an inverted confocal fluorescence microscope (Zeiss LSM 880, Oberkochen, Germany).

### 4.5. RNA Isolation and qPCR

Total mRNA was isolated from cells using TRIzol (Invitrogen, Carlsbad, CA, USA). cDNAs were synthesized from 1 μg of total RNA using HiScript III RT SuperMix for qPCR (Vazyme, Cat#R323-01) (Vazyme, Nanjing, China). qPCR was performed with ChamQ SYBR qPCR Master Mix (Vazyme, Cat#Q341-02) (Vazyme, Nanjing, China) using the Step One Plus Real-Time PCR System (Vazyme, Nanjing, China) according to the manufacturer’s instructions. Relative mRNA expression levels were calculated using the ΔΔCt method. Paired primer sequences for qPCR amplification reactions are listed in Appendix A.

### 4.6. Endothelial Permeability Assay

Cells were seeded into Transwell inserts (pore diameter = 3 μm) at a density of 10^5^ cells/insert, and cultured in a 24-well plate containing 1 mL of complete medium per well for 3 days. Transepithelial electrical resistance was measured to confirm the formation of an endothelial monolayer. We rinsed the inserts and put them into new empty wells. We added 600 μL of 4% BSA into the lower well and added 100 μL of 0.67 mg/mL Evans Blue solution into the insert. After an hour of balance in the cell incubator, 200 μL of solution in each beneath well was collected to measure the absorbance at λ = 620 nm. The concentration of Evans Blue was determined according to a standard curve obtained by serially diluted standard solution.

### 4.7. NO Measurement

NO content was measured with the Griess Reagent System (Promega, Madison, WI, USA, Cat#G2930). We allowed the reagents to equilibrate to room temperature before use. The standard solution and culture medium samples were dispensed into a 96-well plate in triplicate. After the reactions according to the instruction, we measured the absorbance within 30 min with a filter between 520 nm and 550 nm.

### 4.8. ELISA Test for PGI_2_

The test was conducted using an ELISA kit for PGI_2_ (MEIMIAN, Cat# MM-1362H1) (MEIMIAN, Yancheng, China). The kit was balanced to room temperature before use. We prepared a set of concentration gradient solutions of standard substances. We added standard solution and culture medium samples into the wells pre-coated with PGI_2_ antibody and incubated them for 30 min at 37 °C. We washed the plate with cleaning buffer five times. Enzyme-labeled reagent was then added. After 30 min of incubation at 37 °C, the reagent was washed. Chromogenic solution was then added for a 10 min incubation ended using stop buffer. We examined the OD value within 15 min.

### 4.9. Anticoagulation Test

HUVECs were cultured in 24-well plates for three days in advance. Fresh human blood was collected and made into platelet-rich plasma (PRP) with a preparation kit (YuanYe Biotech, Cat#R30057) (YuanYe, Shanghai, China). After removal of the culture medium, 200 μL blood–PRP solution was added into each well with 20 μL of 0.1 M CaCl_2_ as the agonist for blood coagulation. Another 130 μL of PBS was also added to prevent instant coagulation. Blood clotting mass was evaluated after reaction on a shaker at 90 rpm and 37 °C for 30 min.

### 4.10. Statistical Analysis

Data were analyzed using GraphPad Prism software (GraphPad 9.0). Statistical significance was determined by Dunnett’s multiple comparisons test (Figure 1B,C, Figure 2B,C, Figure 3B–D, Figure 4B–F, Figure 5E and Appendix A) or Tukey’s multiple comparisons test (Figure 4H and Figure 5A,B,D). *p* values of less than 0.05 were deemed statistically significant.

## Figures and Tables

**Figure 1 ijms-26-07183-f001:**
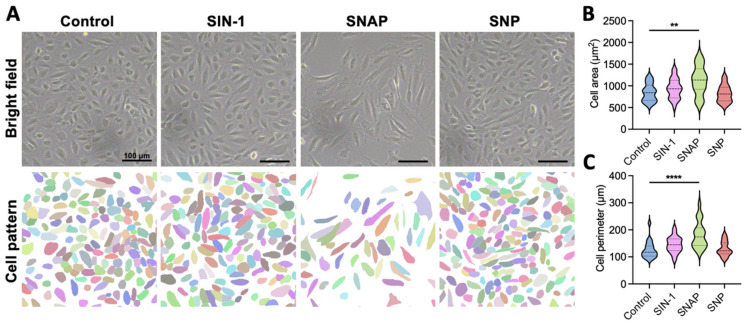
Morphological analysis of HUVECs treated with different nitric oxide donors. (**A**) Representative images and corresponding segmentation of HUVECs cultured under different conditions: 3-morpholinosydnonimine (SIN-1, 0.1 mM), S-nitroso-N-acetylpenicillamine (SNAP, 1 μM), sodium nitroprusside (SNP, 100 μM), and control (PBS). Scale bar: 100 μm. (**B**) Violin plots of average cell area based on segmentation data. Center line shows the median; the other 2 lines indicate upper and lower quartiles, respectively (n = 20 cells from 3 independent experiments; ** *p* < 0.01 vs. control). (**C**) Violin plots of mean cell perimeter based on segmentation data. Center line shows the median; the other 2 lines indicate upper and lower quartiles, respectively (n = 20 cells from 3 independent experiments; **** *p* < 0.0001 vs. control). Data were analyzed using Dunnett’s multiple comparisons test (**B**,**C**).

**Figure 2 ijms-26-07183-f002:**
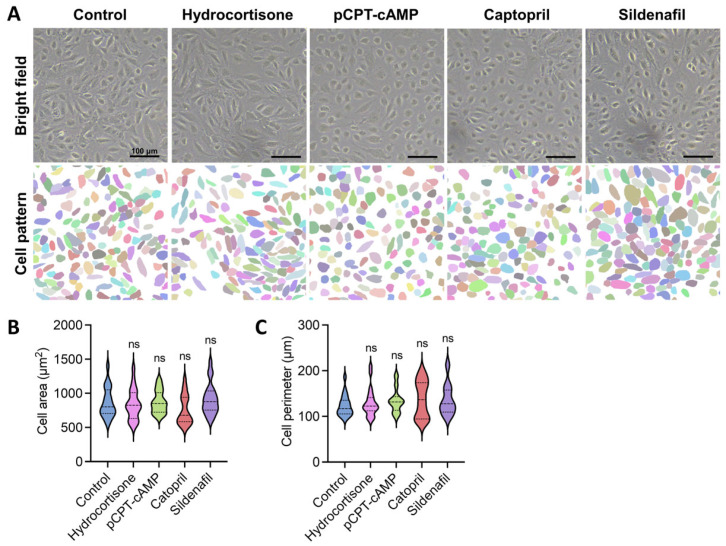
Screening of endothelial cell-activating agents. (**A**) Representative images and corresponding segmentation of HUVECs cultured under different conditions: hydrocortisone (0.1 μM), sildenafil (10 μM), captopril (0.3 mM), and 8-(4-chlorophenylthio) adenosine 3′,5′-cyclic monophosphate sodium salt (pCPT-cAMP, 1 mg/mL). Scale bar: 100 μm. (**B**) Violin plots of average cell perimeter based on segmentation data. Center line shows the median; the other 2 lines indicate upper and lower quartiles, respectively (n = 20 cells from 3 independent experiments; ns: not significant). (**C**) Violin plots of average cell area based on segmentation data. Center line shows the median; the other 2 lines indicate upper and lower quartiles, respectively (n = 20 cells from 3 independent experiments; ns: not significant). Data were analyzed using Dunnett’s multiple comparisons test (**B**,**C**).

**Figure 3 ijms-26-07183-f003:**
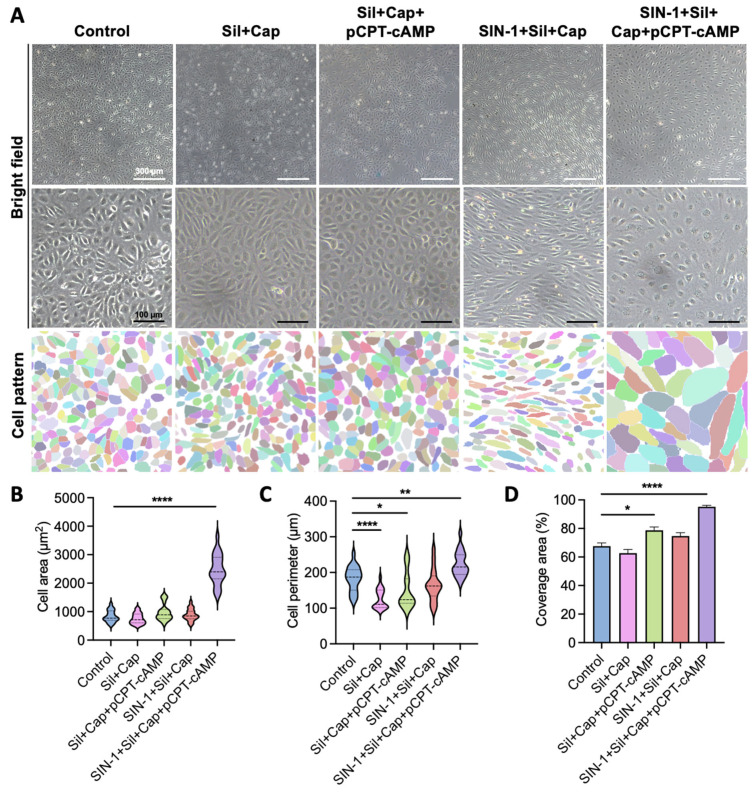
Optimization of agent combination for efficient endothelialization. (**A**) Representative images and corresponding segmentation of HUVECs cultured with standard medium or a selective combination of SIN-1 (100 μM), captopril (0.3 mM), sildenafil (10 μM), and pCPT-cAMP (1 mg/mL). Scale bar: 300 μm (low magnification); 100 μm (high magnification). (**B**) Violin plots of average cell area based on segmentation data. Center line shows the median; the other 2 lines indicate upper and lower quartiles, respectively (n = 20 cells from 3 independent experiments; **** *p* < 0.0001 vs. control). (**C**) Violin plots of average cell perimeter based on segmentation data. Center line shows the median; the other 2 lines indicate upper and lower quartiles, respectively (n = 20 cells from 3 independent experiments; * *p* < 0.05, ** *p* < 0.01, **** *p* < 0.0001 vs. control). (**D**) Quantitative analysis of surface coverage efficiency. Data are presented as mean ± SEM (n = 3 independent experiments; * *p* < 0.05, **** *p* < 0.0001 vs. control). Data were analyzed using Dunnett’s multiple comparisons test (**B**–**D**).

**Figure 4 ijms-26-07183-f004:**
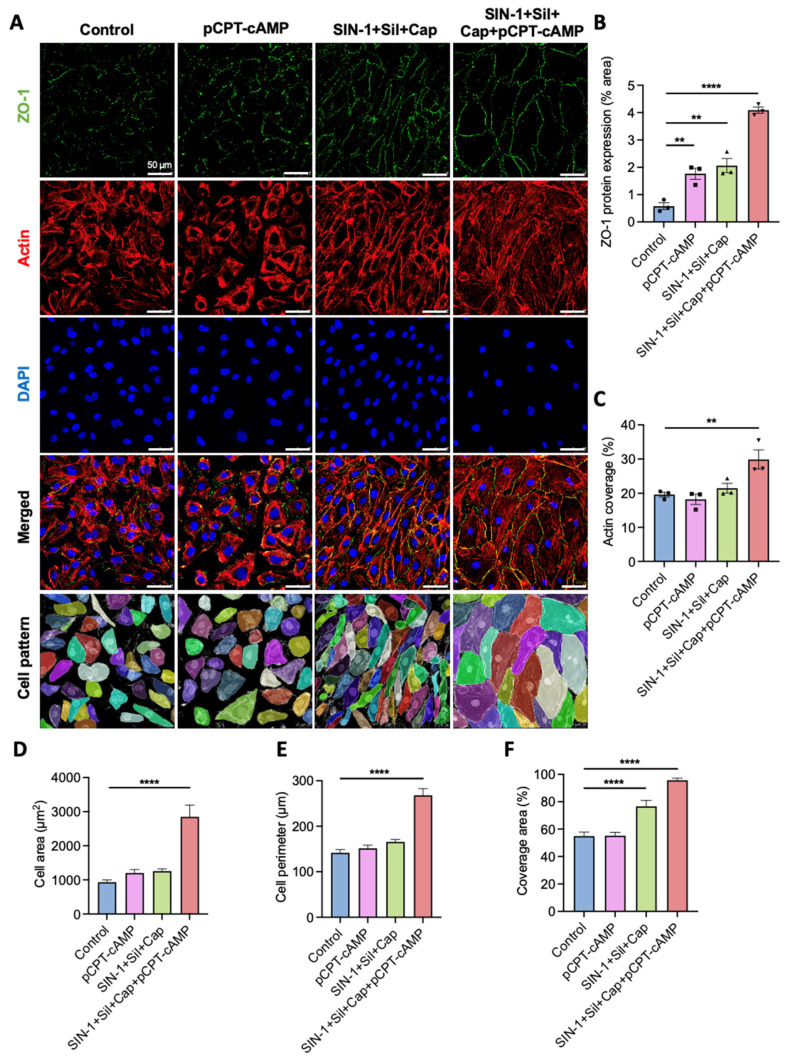
Effects on intercellular junctions and cytoskeletal organization. (**A**) Representative immunofluorescence images of ZO-1 (green) and actin (red) staining with corresponding segmentation. Scale bar: 50 μm. (**B**) Quantitative analysis of ZO-1 expression levels, presented as a percentage of the positive area. Data are presented as mean ± SEM (n = 3 biological replicates; ** *p* < 0.01, **** *p* < 0.0001 vs. control). (**C**) Quantitative analysis of actin coverage, presented as a percentage of the positive area. Data are presented as mean ± SEM (n = 3 biological replicates; ** *p* < 0.01 vs. control). (**D**) Quantitative analysis of average cell area. Data are presented as mean ± SEM (n = 3 biological replicates; **** *p* < 0.0001 vs. control). (**E**) Quantitative analysis of average cell perimeter. Data are presented as mean ± SEM (n = 3 biological replicates; **** *p* < 0.0001 vs. control). (**F**) Quantitative analysis of surface coverage efficiency. Data are presented as mean ± SEM (n = 3 biological replicates; **** *p* < 0.0001 vs. control). (**G**) Transcriptional levels of specific tight junction-associated molecules modulated by the four-component combination and individual drugs. Data are presented as mean ± SEM (n = 3 biological replicates). (**H**) Vascular endothelial permeability examined by transudatory Evans Blue. Data are presented as mean ± SEM (n = 3 biological replicates; **** *p* < 0.0001). (**I**) Relative gene expression of key regulators in associated signaling pathways (color indicator: a gradient ramp of blue–gray–pink representing values of 0–1–300, respectively). Data were analyzed using Dunnett’s multiple comparisons test (**B**–**F**) and Tukey’s multiple comparisons test (**H**).

**Figure 5 ijms-26-07183-f005:**
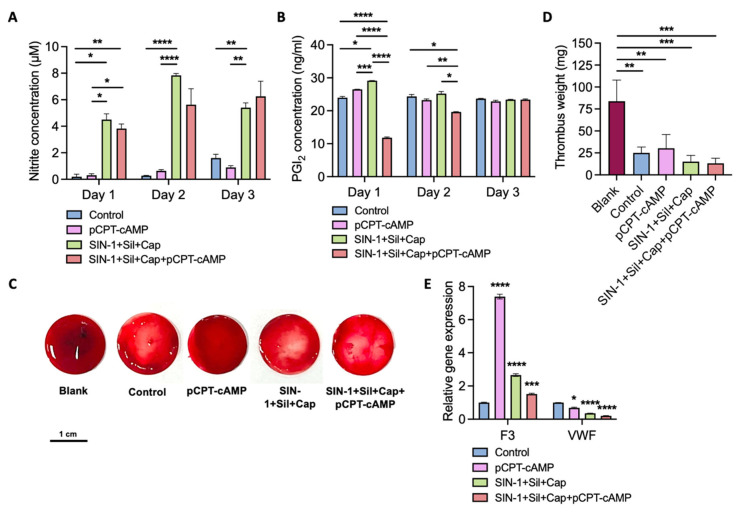
Anticoagulative properties of conditioned endothelial monolayers. (**A**) NO content in culture supernatants over three days. Data are represented by mean ± SEM (n = 3 biological replicates; * *p* < 0.05, ** *p* < 0.01, **** *p* < 0.0001). (**B**) PGI_2_ content in culture supernatants over three days. Data are represented by mean ± SEM (n = 3 biological replicates; * *p* < 0.05, ** *p* < 0.01, *** *p* < 0.001, **** *p* < 0.0001). (**C**) Representative images of anticoagulation assays using platelet-rich plasma. Scale bar: 1 cm. Human full blood–PRP-CaCl_2_ solution was prepared for the anticoagulation test. Thrombus mass was measured after 30 min of reaction. (**D**) Quantitative analysis of thrombus mass. Data are represented by mean ± SEM (n = 3 biological replicates; ** *p* < 0.01, *** *p* < 0.001). (**E**) Relative gene expression of F3 and VWF in HUVECs. Data are represented by mean ± SEM (n = 3 biological replicates; * *p* < 0.05, *** *p* < 0.001, **** *p* < 0.0001 vs. control). Data were analyzed using Tukey’s multiple comparisons test (**A**,**B**,**D**) and Dunnett’s multiple comparisons test (**E**).

**Table 1 ijms-26-07183-t001:** Reagents and antibodies.

Reagent or Antibody	Source	Identifier
SNP	KeyGen BioTECH (Nanjing, China)	Cat#KGR0017
SIN-1	Cayman Chemical (Ann Arbor, MI, USA)	Cat#82220
SNAP	Cayman Chemical (Ann Arbor, MI, USA)	Cat#82250
Hydrocortisone	MCE (Monmouth Junction, NJ, USA)	Cat#HY-N0583
Captopril	Sigma-Aldrich (St. Louis, MO, USA)	Cat#C8856
Sildenafil	Solarbio (Beijing, China)	Cat#6070
pCPT-cAMP	Sigma-Aldrich (St. Louis, MO, USA)	Cat#C3912
Dylight-554 Phalloidin	CST (Danvers, MA, USA)	Cat#13054S
ZO-1 antibody	Abcam (Cambridge, MA, USA)	Cat#ab221547

## Data Availability

The original contributions presented in this study are included in the article and Appendix A. Further inquiries can be directed to the corresponding author.

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
