# Peer review of "Combined Pharmacological Conditioning of Endothelial Cells for Improved Vascular Graft Endothelialization"

_ijms, 2025, doi:10.3390/ijms26157183_

Round 1

Reviewer 1 Report

Comments and Suggestions for Authors

attached file (pdf)

Reviewer 2 Report

Comments and Suggestions for Authors

In this manuscript, Lu et al. demonstrate the synergistic effect of the combination of pharmacological compounds on endothelialization while maintaining antithrombotic effects. The manuscript is well written, and the hypothesis is original and relevant, which is important for advancing our understanding of the endothelialization process. The experimental results support the overall interpretation. However, the manuscript lacks mechanistic detail. I have outlined some suggestions below.

  1. This manuscript lacks data evaluating how the combination of pharmacological compounds (SIN-1 + sildenafil + captopril + pCPT-cAMP) affects cell cytotoxicity and cell death compared to treatment with individual compounds or an untreated control in HUVEC cells, which is essential because it can influence the observed results.
  2. The authors should provide research evidence on whether this combination affects the overall expression of specific tight junction-associated proteins (ZO-1, Claudins, JAMs), compared to individual compounds, using immunoblotting. It will also clarify whether only the expression of ZO-1 protein was affected or if other components of the tight junction were also impacted.
  3. This manuscript can be further improved with data on downstream signaling mechanisms, showing how these compounds work together to activate intracellular signaling pathways that modulate cell morphology and tight junction formation compared to individual compounds or untreated cells. This data will help understand the effects and rationale behind combining these compounds.

Round 2

Reviewer 2 Report

Comments and Suggestions for Authors

The authors have addressed most of the concerns raised during the review process. The manuscript has significantly improved.

Author Response

Comment:

The authors have addressed most of the concerns raised during the review process. The manuscript has significantly improved.

Answer:

Thank you for your time helping us improve the manuscript. Your inspiring suggestions have been of great value to us.